# Influence of Internal and External Foot Rotation on Peak Knee Adduction Moments and Ankle Moments during Gait in Individuals with Knee Osteoarthritis: A Cross-Sectional Study

**DOI:** 10.3390/bioengineering11070696

**Published:** 2024-07-09

**Authors:** Yongwook Kim

**Affiliations:** Department of Physical Therapy, College of Medical Sciences, Jeonju University, 303 Cheonjam-ro, Wansan-gu, Jeonju 55069, Republic of Korea; ptkim@jj.ac.kr

**Keywords:** adduction moment, foot progression angle, knee joint, osteoarthritis

## Abstract

The aim of the study was to verify the effects of foot progression angle (FPA) modification during walking on the internal moments of the ankle and knee joints in individuals with knee osteoarthritis (OA). Biomechanical changes such as increased knee adduction moment (KAM) during walking are known to be involved in the development and severity of knee OA. Although various FPA modifications during gait have been applied to reduce peak KAM, few studies have investigated the effects of applying toe-in or toe-out walking modifications for knee OA on peak KAM and three-dimensional (3D) moments of the ankle joint. Kinetic moment variables were acquired from 35 individuals with medial knee compartment OA. A 3D motion analysis system and two force platforms were used to acquire KAM and 3D moments of both ankle joints during gait. Visual3D was used to obtain final moment data for statistical processing. Repeated-measures analysis of variance with Bonferroni adjustment was used to compare kinetic and kinematic values for each FPA walking condition. There was a significant decrease (*p* < 0.01) in first peak KAM when walking with an internal rotation foot position compared to normal foot position walking. Also, there was a significant decrease (*p* < 0.01) in second peak KAM when walking with an external rotation foot position compared to normal foot position walking. Compared to a normal foot position, peak ankle inversion moment of the external rotation foot position walking showed a significant decrease (*p* < 0.05). There were no interactive effects between FPA condition and limb sides for any KAM values (*p* > 0.05). The results showed no significant increase in the ankle joint moment value during gait for FPA modification conditions. Thus, the clinical implications of this study suggest that modification of the FPA in patients with OA to reduce KAM does not negatively impact the 3D ankle moments.

## 1. Introduction

Knee osteoarthritis (OA), one of the degenerative diseases, is rapidly increasing due to recent increases in human average lifespan and an aging society [1,2]. Knee OA is a common disease that occurs in about one-third of people over the age of 60, with a prevalence rate of 20.2% in men and 50.1% in women [3]. Biomechanical changes such as increases of peak knee adduction moment (KAM) and knee varus shear force during walking are known to be involved in the development and severity of knee OA [4,5,6]. Because it is generally difficult to measure joint loads directly, joint moments such as KAM are often used as alternative measures of the force applied to a joint [7]. Clinically, KAM is used for diagnosing knee OA and predicting knee pain, knee deformity, and the degree of knee OA progression [8,9]. KAM is a coronal kinetic variable. It typically has two distinct peaks during the stance phase of gait. The first peak occurs around the 25% stance phase during walking. The second peak elicits around the 75% stance phase [3]. The first peak KAM is typically greater than the second one. It is larger for patients with medial knee OA than for age-matched healthy individuals [10]. Peak KAMs during walking are determined by various factors such as gait pattern, knee pain avoidance strategy, and foot progression angle (FPA). Abnormally increased KAM during gait is closely related to worsening of musculoskeletal diseases such as knee OA [4,11,12].

From a biomechanical perspective, changes in kinematic and kinetic variables of any joints or segments during gait may affect kinesiologic variables of all other connected joints and segments of the musculoskeletal system [13,14]. Therefore, changes in FPA modification conditions such as internal rotation or external rotation of the foot in contact with the floor during the stance phase of gait may affect the KAM [6,14,15,16,17,18]. A previous study reported that the second peak KAM was significantly decreased during toe-out walking compared to the baseline gait [6]. Additionally, the first peak KAM in internal rotation of FPA during gait is significantly decreased in patients with medial knee OA [19]. In a study on the second peak KAM, both the 10° toe-in gait and the baseline gait showed the same moment value (−0.46 Nm/kg) without a significant difference [7]. In addition, there were no significant differences between the two FPA conditions on the second peak KAM during toe-in gait and baseline gait in patients with knee OA [19]. On the other hand, Lynn et al. [20] reported a significantly greater KAM value of the late stance in internal foot position walking than in normal walking. Although many previous studies have reported that the FPA modifications can reduce first and second peak KAMs in patients with medial knee OA, the peak KAM change varies significantly between individuals in response to the achievement of the target FPA, anatomical variations, and biomechanical influence of the musculoskeletal connection of the lower extremities [5]. Therefore, evaluating objective and quantitative peak KAMs according to changes in FPA during gait is important for obtaining positive clinical outcomes of patients with knee OA.

Studies verifying effects of foot rotation modifications during gait have mainly focused on the effects of FPA during gait on the biomechanics of the knee joint [7,21]. There is a need to evaluate the effects of these modifications on other lower extremity joints. Most of the related studies have mainly examined effects of gait modification on KAM and biomechanical values of the hip joint [7,22,23]. However, there have been few studies on the effects of the foot modification angle during walking on kinematics and kinetics of the ankle joint along with its effect on KAM. Although it is not known how FPA changes affect the ankle joint, increased ankle joint moments in three dimensional planes of motion may increase biomechanical loads on soft tissues around the ankle joint and increase the incidence of musculoskeletal disorders such as ankle OA [24]. Therefore, the purpose of this study was to investigate the effects of different FPA conditions during free walking on the kinetics of the ankle and knee joints in pain-free individuals with knee osteoarthritis using a 3D motion analysis system with two force platforms.

## 2. Materials and Methods

### 2.1. Study Participants

This study was conducted with a cross-sectional study design. A sample size was calculated with G*Power version 3.1.9.2 software based on variables for the second peak KAM during FPA modification gait retraining in individuals with medial knee OA [25]. The sample size obtained using this study data was 23 people, with an effect size of 1.0, an alpha level of 0.05, and 90% power. Thirty-five participants (9 males and 26 females) diagnosed with knee OA participated in this study. All participants did not experience any knee pain while completing simple, daily activities such as walking. However, they reported difficulties and discomfort during walking up and down stairs, and standing up from the floor due to knee OA. Their average Kellgren–Lawrence classification was 1.8 ± 0.58 (Grade I: *n* = 10, Grade II: *n* = 22, Grade III: *n* = 3). The Kellgren–Lawrence classification is a common method of classifying the severity of knee OA using five grades [26]. Most of the participants in this study were grade I (doubtful joint space narrowing and possible osteophytic lipping), grade II (definite osteophytes and possible joint space narrowing), or grade III (moderate multiple osteophytes, definite narrowing of joint space and some sclerosis). Participants were diagnosed by an orthopedic specialist and recruited from a local orthopedic clinic. The mean illness duration of knee OA after initial diagnosis was 5.2 ± 2.4 years. Inclusion criteria were the following: (1) no surgical history of knee or ankle joints; (2) no severe pain of the lower extremity joints or segments; (3) Kellgren–Lawrence classification 3 or less; and (4) able to walk free and unaided according to instructions. If participants had neurologic conditions affecting walking, painful knee OA, or systemic rheumatoid arthritis, they were excluded from this study. An experienced researcher fully explained the research purpose and experimental procedures of this study to participants. All participants voluntarily agreed to participate in this study. Written informed consents were obtained from all participants. The study assessment protocol and design were approved by the Institutional Review Board of Jeonju University (jjIRB-200714-HR-2020-0708).

### 2.2. Instrumentation for Gait Analysis and Data Acquisition

Biomechanical data during gait were acquired from the motion analysis experimental laboratory at Jeonju University. A 3D motion analysis system (Vicon Inc., Oxford, UK) based on eight infrared cameras and an 8 m walking pathway was used to acquire 3D moments of both ankle joints and coronal knee joint moments during gait (Figure 1). The sampling rate of the motion capture camera system was set at 100 Hz. The lab had two force platforms (AMTI, Watertown, MA, USA) for sampling data at 500 Hz. They were placed in the center of the walking way to obtain the 3D ground reaction forces needed for the determination of knee and ankle moments for different FPA modification conditions.

A T-frame bar (0.75 m) was used to recognize the origin of the 3D spatial coordinates of the laboratory and for initial calibration of the object and motion analysis system. Analog kinetic and kinematic data obtained from two force platforms and eight capture cameras were converted to digital data and processed through the Nexus program (version 1.8.5, Vicon Inc., Oxford, UK) on the main computer. Anatomical labeling of reflective markers and a lower limb segment model, which was the basis for gait analysis, were created through the Nexus program [27]. In addition, the final c3d file for each gait trial was created through integrated processing of all kinetic and kinematic data. Visual3D (C-Motion Inc., Germantown, MD, USA) was used with the final c3d data exported from the Nexus program to obtain final quantitative moment data for statistical processing and 3D graphic results of knee and ankle moments. Visual3D version 6 software created a virtual skeletal model for each subject based on anatomical markers and tracking markers and was used to analyze kinematic data using the Calibrated Anatomical System Technique (Figure 2) [28]. The default Visual3D lower-body model was selected as the most appropriate model and marker set for the biomechanical analysis of this study. Kinematic data were low-pass filtered with a 4th order Butterworth filter with a cut-off frequency of 6 Hz. Kinetic data were low-pass filtered using a 4th order Butterworth filter with a cut-off frequency of 15 Hz [29]. The kinematic model created in Visual 3D was used to quantify the motion at the lower extremity joints with rotations being expressed relative to the static trial. The X-Y-Z Cardan angle sequence was used to calculate joint angles and a standard inverse dynamics analysis was conducted to synthesize the marker trajectory and ground reaction force data for internal moment estimation. All moment data of the knee and ankle joints were normalized to mass and analysis time of the moment normalized to 100% of the stance phase during gait. Analysis of knee moments and foot angles for each of the three FPA walking trials was carried out.

Ten reflective markers (14 mm) were attached bilaterally on the medial and lateral malleoli, medial and lateral femoral epicondyles, and greater trochanters to create joint positions (Figure 3). Four reflective markers were attached to the anterior superior iliac spine and posterior superior iliac spine on both sides to form the pelvic segment. Four cluster markers were attached bilaterally on the lower leg and thigh segments according to the six degrees of freedom (6DOF) model to present lower skeletal segments [29]. Additionally, five reflective markers were attached to foot segments to measure FPA. Foot markers were attached onto the dorsal side of the first and fifth metatarsal heads, the dorsal center of the midfoot, and the medial and lateral calcaneus. The FPA was calculated as the angle between the forward direction of the laboratory coordinate system and the line connecting the first metatarsal head and the medial calcaneus marker positions during 15–40% of stance [7]. After setting the marker, static calibration was executed to create the laboratory coordinate system for the motion analysis system and force platforms. The average FPA value in each walking condition was calculated through 100% of stance phase.

### 2.3. Gait Analysis

The experimental protocol for gait analysis included three different FPA modification trials: normal foot position (NFP), maximal possible internal rotation foot position (IFP), and maximal possible external foot position (EFP). To maintain a consistent foot progression angle in each FPA walking trail, the foot rotation when each subject walked naturally was set as the baseline NFP. IFP and EFP conditions were then set by adding or subtracting about 20° to this angle [20]. Additionally, to ensure consistent walking speed and step width, a tape line was attached to the walking pathway so that each subject could know the degree of FPA rotation achieved during the walking practice. For each walking condition (NFP, IFP, and EFP), a total of 8–10 walking trials that achieved the target FPA line were analyzed. The gait cycle was determined from heel strike to the next heel strike based on one lower limb. Each gait trial consisted of an average of 2.5 useful gait cycles. An average of 25 gait cycles for each FPA condition was used in the final analysis. To fill the gaps in the marker trajectories, the spline fill and pattern fill methods of the Nexus program were used. Gait cycles containing gaps that could not be restored by these methods were excluded from the final analysis. The participants were asked to walk wearing a standardized shoe provided by an experimenter along an 8 m walkway at their self-selected speed. Kinetic and kinematic data were calculated for every walking trial and averaged over each FPA gait condition. If the trial was not completed successfully in an FPA condition, an additional walking trial was performed. The experimental order of each FPA condition was randomly assigned.

### 2.4. Statistical Analysis

Descriptive analysis was performed to determine general characteristics such as height and weight of the participants, as well as moment mean and standard deviation of the ankle and knee joints. Knee and ankle moments were normalized by subject weight [7]. Knee moment values were compared between main FPA conditions: 1st peak KAM during 0–50% of stance phase, peak KAM at mid stance, and 2nd peak KAM during 50–100% of stance phase. For comparative analysis of ankle moments for each FPA walking condition, characteristic peak moment variables generated in each 3D motion plane were used. Peak ankle moment variables used in the analysis were dorsiflexion, plantar flexion, inversion, eversion, internal rotation, and external rotation moment that occurred in a specific period of stance phase. Because all analysis variables satisfied the parametric test, the Kolmogorov–Smirnov test used to verify normal distribution with two-way repeated-measures analysis of variance (ANOVA) with Bonferroni adjustment was used to compare biomechanical values for each FPA walking condition. If the significance of the main effect (FPA condition or limb side) was confirmed in the ANOVA, pairwise comparisons were performed using post hoc tests. All knee and ankle moment data used in the ANOVA analysis were obtained from both lower extremities. To explain the effect size between independent and dependent variables, partial η2 between FPA conditions and lower limb sides was analyzed. If the partial η2 was 0.01, the effect size was small, if it was 0.06, it was medium, and if it was 0.14 or higher, the effect size was large [4]. All analyses were performed using SPSS version 26 (IBM Corp, Armonk, NY, USA). Statistical significance was determined at an α level of 0.05.

## 3. Results

There were no significant differences in walking speed, cadence, or step length between FPA walking conditions (all *p* > 0.05) (Table 1).

### 3.1. Peak Knee Moments Occurring in Coronal Motion Plane

All knee joint moment data required for analysis satisfied Mauchly’s sphericity assumption. The first peak KAM (F = 9.753, *p* = 0.004) and the second peak KAM (F = 27.624, *p* = 0.00006) were significantly different according to FPA walking trials (Table 2). The first peak KAM of the IFP walking condition showed a significant decrease compared to that of the NFP condition (*p* < 0.01), and the partial η2 as the effect size of the first peak KAM according to FPA conditions was 0.17 (Table 3, Figure 4). Additionally, the second peak KAM of the EFP condition showed a significant decrease compared to that of the NFP condition (*p* < 0.01), and the partial η2 as the effect size of the second peak KAM according to FPA conditions was 0.30 (Table 3, Figure 4). There were no interactive effects between FPA condition and knee sides for any KAM values (*p* > 0.05), and the partial η2 as the effect size of the peak KAMs according to FPA conditions and knee sides was 0.01 (Table 2).

### 3.2. Moment Results of Ankle Joint According to FPA Conditions

There was a significant difference only in the peak inversion moment generated at the ankle joint in the stance phase during walking depending on FPA conditions (F = 6.107, *p* = 0.019), and the partial η2 as the effect size of the ankle inversion moment according to FPA conditions was 0.11 (Table 4). Compared to the NFP condition, the peak inversion moment of the EFP walking condition generated at 50–75% stance phase showed a significantly decreased moment (*p* = 0.034, 95% CI = −0.07~−0.01) (Table 5). On the other hand, except for the inversion peak moment value, there were no significant differences in other ankle peak moment values according to FPA walking conditions (*p* > 0.05).

## 4. Discussion

This study was conducted to determine effects of gait modifications such as IFP and EFP walking on knee KAM and 3D moment characteristics occurring at the ankle joint during the stance phase in patients with knee OA. Increased KAM during walking not only increases the loading force imposed on the medial compartment of the knee joint, but also promotes the development of clinical symptoms of knee OA such as pain and inflammation [30,31]. Therefore, various gait modification methods such as medial knee thrust, lateral trunk lean, and toe-in or toe-out gait have been applied to patients with knee OA to reduce KAM and alleviate clinical symptoms [16,32,33,34]. Comparison of effects between gait modifications in previous studies is limited by diverse sample characteristics and difficulties in consistently applying gait modifications [32,35]. To address these methodological limitations of previous studies, this study clearly established inclusion criteria for study participants with knee osteoarthritis through pain severity and clinical evaluation as the Kellgren–Lawrence test. Additionally, guidelines were set for the walking pathway so that three different FPA walking conditions were applied as equally and consistently as possible to all participants.

This study analyzed how changes in FPA conditions affected knee joint and ankle joint moments during walking in 35 patients with medial knee osteoarthritis. Results of this study showed that the first peak value of KAM in the IFP walking condition was significantly decreased by 19.1% compared to that of NFP as the baseline walking condition. Similar to previous studies, our results demonstrated the clinical effectiveness of specific gait modifications in reducing knee joint discomfort during walking, in which participants who have a choice of different gait modifications preferred IFP gait the most [32,34,36]. In general, the first peak KAM was greater than the second. The results of this study demonstrated a significant reduction in the first peak of the KAM during IFP walking. The reason why the FPA condition of IFP gait significantly reduced the first peak of the KAM may be due to movement of the loading center to the lateral side as the knee joint central axis moved medially [19]. The second peak of the KAM (−0.50 Nm/kg) of the IFP gait tended to increase compared to that of the NFP condition (−0.44 Nm/kg), although no significant difference was shown. 

In contrast to IFP gait that showed a significant decrease in the first peak KAM, the EFP gait showed a significant reduction in the second peak KAM compared to the NFP condition in individuals with knee OA. However, there was no significant difference in the first peak KAM value between NFP (−0.42 Nm/kg) and EFP (−0.47 Nm/kg) gait conditions. These results were similar to previous studies reporting reduction of the second peak KAM during EFP gait in patients with knee OA [6,19,37]. The current study applied an EFP walking trial, which was increased by 20° compared to the NFP angle. The characteristics of the present study participants were also individuals with knee osteoarthritis, not healthy participants. Peak KAM values of the present study for the EFP walking condition were somewhat different from previous studies [38,39,40,41]. These differences might be attributed to different characteristics of study participants, clinical severity of knee OA, and variations in study design. Specifically, most participants of the present study were patients with mild knee OA.

Our results showed no significant increase in most ankle joint moments in the stance phase during walking for both 20° IFP and 20° EFP modifications. Therefore, the clinical implication of this study is that FPA modification gait training to manage KAM in individuals with knee OA can generally be performed without altering ankle moments. However, in some participants, the IFP gait modification increased the peak ankle adduction moment when ankle joint contact forces peaked. This finding of altered ankle adduction moments suggests that for patients with musculoskeletal deficits of the ankle joint such as ankle OA, the effectiveness of IFP or EFP walking trials at the ankle joint should be verified case by case before adopting FPA gait to manage the KAM. Although IFP gait trials showed a significant decrease in the first peak KAM with a significantly decreased second peak KAM in EFP walking, clinical application of the FPA walking modification for patients with knee osteoarthritis requires a cautious approach. Because the first peak KAM is more closely related to current clinical symptoms, severity, and prognosis of knee OA than the second peak KAM, the decrease in the first peak KAM is considered more important [19]. However, considering the impact of a 1% increase in KAM on knee OA symptoms [42], it will not be possible to overlook the tendency to increase the first peak KAM during the EFP walking condition as well as the tendency to increase the second peak KAM during IFP walking.

The current study has some strengths. Previous studies related to KAM and FPA modifications investigated effects of foot internal or external rotation conditions on kinematics and kinetics of the knee joint itself during gait [5,6,19,20,25]. This study was performed to determine the effects on two peak KAMs and 3D moments of the ankle joint using three different FPA modifications in individuals with knee OA through a reliable and objective 3D motion analysis system and force platforms. The present study showed that in participants with mild or moderate knee OA, FPA walking modifications affected peak KAMs without affecting the increase in peak ankle moment. Therefore, clinical application of FPA modification should be approached cautiously through quantitative analysis of the two peak KAMs in patients with knee osteoarthritis. Although abnormally high, the peak KAM is a risk factor for medial compartment knee OA; abnormally reduced peak KAM might also be a risk factor for lateral compartment knee OA [20].

The limitations of this study are as follows. Because most participants had mild knee OA without severe knee pain or clinical symptoms, results of this study cannot be applied to all knee OA patients. Due to difficulties in recruiting research participants, the experiment could not be conducted with a sufficient number of participants, and there were significantly more female knee OA participants because of the clinical characteristics of the OA disease where it is known to occur more in women than men. We were unable to evaluate skeletal alignment, such as hip anteversion or retroversion which may affect the moment values of lower extremity joints during gait. In addition, participants were not screened or controlled for foot structures that might contribute to excessive rearfoot and midfoot pronation or supination according to the foot progression angle during gait. Various research methodological techniques were used to apply consistent FPA walking modifications. However, there were difficulties with consistency in some walking trials. Future research will need to verify the impact of clinical application of walking aids such as foot and ankle orthoses or insoles and integrated intervention of gait modification retraining on the first and second peak KAMs and the biomechanics of lower extremity joints in a large number of patients with knee OA.

## 5. Conclusions

In conclusion, this study showed that IFP walking modification reduced the first peak KAM and that EFP walking reduced the second peak KAM. Additionally, any modification in FPA did not significantly change ankle moments. However, the impact of this FPA walking retraining intervention on other types of knee musculoskeletal disorders such as lateral compartment knee OA cannot be overlooked. Therefore, clinical application of IFP or EFP gait modification to patients with knee osteoarthritis should be performed based on biomechanical evaluations such as peak KAM. It is also necessary to consider additional applications of foot and ankle orthoses or modified insoles, which have a positive impact on clinical intervention.

## Figures and Tables

**Figure 1 bioengineering-11-00696-f001:**
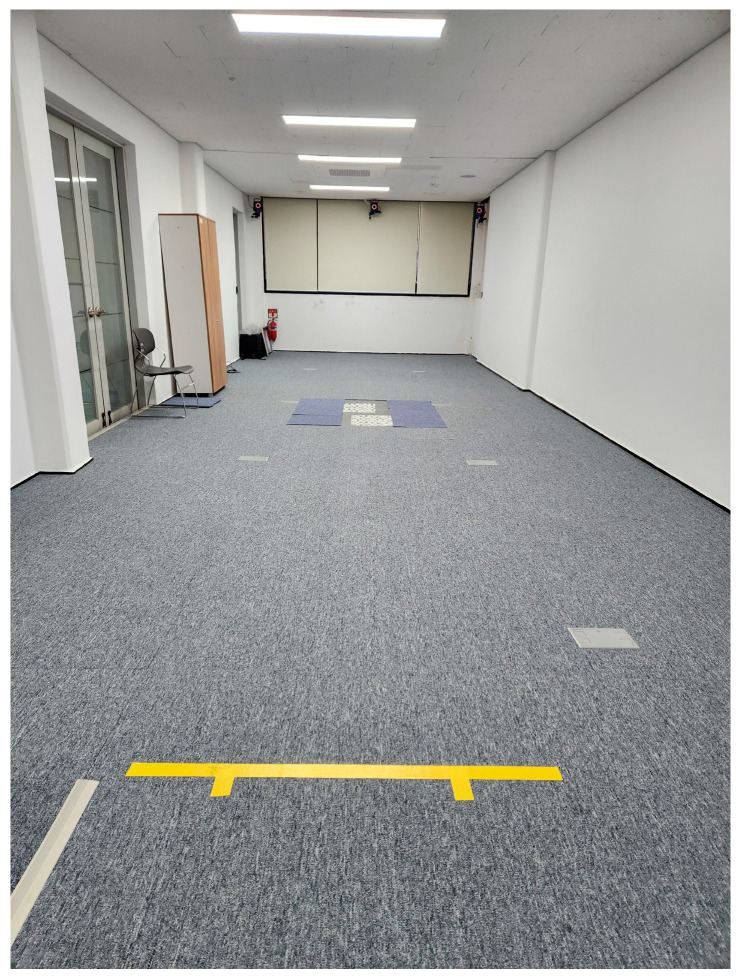
Motion analysis laboratory equipped with 3D motion capture cameras and two force platforms embedded in the center of the experimental 8 m walking pathway.

**Figure 2 bioengineering-11-00696-f002:**
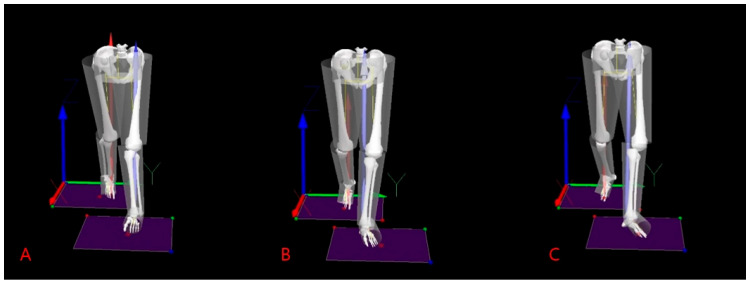
Visual3D representation of virtual musculoskeletal model according to three different foot progression angles in each walking trial. (**A**): Internal rotation foot position; (**B**): Normal foot position; (**C**): External rotation foot position.

**Figure 3 bioengineering-11-00696-f003:**
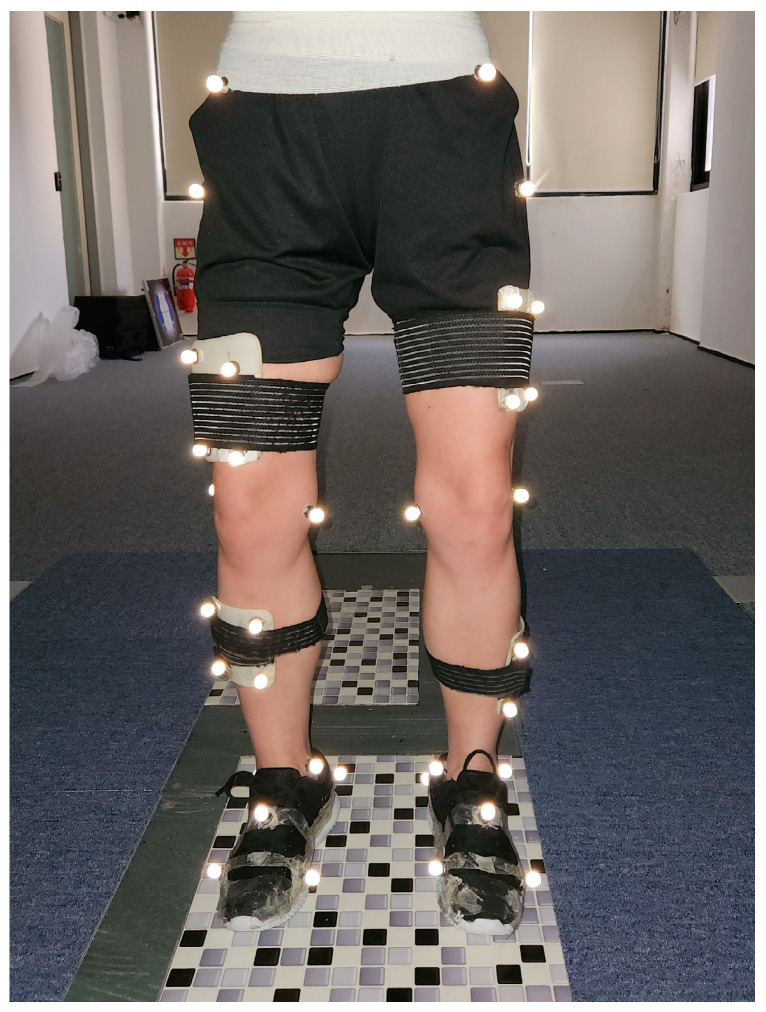
Forty reflective markers set to obtain the static calibration skeletal model and kinetic moment data.

**Figure 4 bioengineering-11-00696-f004:**
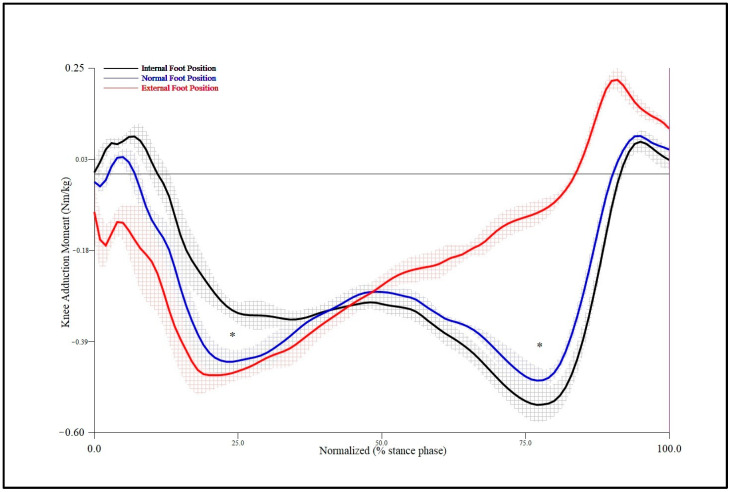
Mean and standard deviation of knee adduction moment for foot progression angle conditions in stance phase during gait. *, Significant main effect for foot progression angle position at *p* < 0.01.

**Table 1 bioengineering-11-00696-t001:** General characteristics of study participants (standard deviation (SD) is presented in parentheses).

Characteristics (N = 35)	Mean (SD)
Age (yr)	61.2 (2.1)
Height (cm)	159.3 (9.5)
Mass (kg)	64.8 (10.2)
Gender	Male: 9, Female: 26
Hip rotation angle (deg)	IFP: −4.49 (2.72), EFP: 11.39 (4.25)
Knee rotation angle (deg)	IFP: −0.61 (1.28), EFP: 1.07 (2.63)
Ankle rotation angle (deg)	IFP: −1.57 (3.47), EFP: 0.65 (2.84)
Foot progression angle (deg)	NFP: 17.84 (7.12), IFP: −10.09 (5.55), EFP: 42.76 (6.80)
Velocity (m/s)	NFP: 1.19 (0.13), IFP: 1.16 (0.13), EFP: 1.20 (0.14)

NFP: normal foot position; IFP: internal foot position; EFP: external foot position. The hip and knee rotation angle values were calculated based on the NFP baseline of the right side. The negative values of joint angle inferred the absolute angle values of internal rotation direction.

**Table 2 bioengineering-11-00696-t002:** Comparisons of peak knee moments of coronal plane by foot progression angle and knee side in stance phase during gait.

Moment Values	Level	F	*p*-Value
Knee adduction moment at first peak 0–25% stance (Nm/kg)	Foot positions	9.753	0.003
	Knee sides	1.375	0.218
	Positions × sides	2.540	0.101
Knee adduction moment at mid stance 25–70% stance (Nm/kg)	Foot positions	0.984	0.326
	Knee sides	0.077	0.734
	Positions × sides	0.183	0.657
Knee adduction moment at second peak 75–100% stance (Nm/kg)	Foot positions	27.624	0.000
	Knee sides	1.779	0.166
	Positions × sides	1.343	0.255

F: represents the ratio of the variance between the groups to the variance within the groups. Knee sides mean between the right and left knees.

**Table 3 bioengineering-11-00696-t003:** Mean (SD) values of knee adduction moments presented for three foot progression angle conditions in stance phase during gait.

Moment Values (Nm/kg)	NFP	IFP	EFP	*p*-Value
1st peak knee adduction moment at around 25% stance	−0.42 * (0.11)	−0.34 (0.08)	−0.47 (0.10)	0.003
Knee adduction moment at mid 50% stance	−0.27 (0.09)	−0.30 (0.08)	−0.27 (0.09)	0.326
2nd peak knee adduction moment at around 75% stance	−0.44 ^†^ (0.14)	−0.50 (0.16)	−0.11 (0.11)	0.000

* Significant difference between NFP and IFP conditions, *p* < 0.01. ^†^ Significant difference between NFP and EFP conditions, *p* < 0.01. NFP: normal foot position; IFP: internal foot position; EFP: external foot position. The negative values inferred the absolute moment values of the knee adduction.

**Table 4 bioengineering-11-00696-t004:** Repeated measures ANOVA comparing three-dimensional ankle movement by foot progression angle conditions and foot side during gait.

Ankle Moment (Nm/kg)	Level	F	*p*-Value
Dorsiflexion moment peak 0–50% stance	Foot position	0.761	0.486
	Ankle sides	1.122	0.299
	Positions × sides	2.744	0.078
Plantar flexion moment peak 50–100% stance	Foot position	0.529	0.571
	Ankle sides	1.161	0.254
	Positions × sides	0.188	0.815
Inversion moment peak 50–75% stance	Foot position	6.107	0.019
	Ankle sides	1.230	0.281
	Positions × sides	0.019	0.843
Eversion moment peak 0–50% stance	Foot position	2.347	0.102
	Ankle sides	0.294	0.758
	Positions × sides	1.003	0.359
Internal rotation moment peak 0–50% stance	Foot position	0.888	0.390
	Ankle sides	0.554	0.568
	Positions × sides	0.290	0.692
External rotation moment peak 50–100% stance	Foot position	1.453	0.197
	Ankle sides	0.482	0.475
	Positions × sides	2.111	0.120

F: represents the ratio of the variance between the groups to the variance within the groups. Ankle sides mean between the right and left ankles.

**Table 5 bioengineering-11-00696-t005:** Mean (SD) ankle moment values presented for all three foot progression conditions in stance phase during gait.

Ankle Moment (Nm/kg)	NFP	IFP	EFP
Dorsiflexion peak 0–50% stance	0.18 (0.08)	0.18 (0.09)	0.17 (0.09)
Plantarflexion peak 50–100% stance	−1.34 (0.26)	−1.34 (0.27)	−1.37 (0.26)
Inversion peak 50–75% stance	−0.09 * (0.04)	−0.11 (0.06)	−0.04 (0.05)
Eversion peak 0–50% stance	0.32 (0.26)	0.32 (0.25)	0.30 (0.26)
Internal rotation peak 0–50% stance	0.05 (0.08)	0.05 (0.07)	0.03 (0.05)
External rotation peak 50–100% stance	−0.24 (0.16)	−0.27 (0.19)	0.23 (0.16)

* A significant difference between NFP and EFP conditions, *p* = 0.019. NFP: normal foot position; IFP: internal foot position; EFP: external foot position.

## Data Availability

The original contributions presented in the study are included in the article, further inquiries can be directed to the corresponding author.

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
