# Peer review of "Influence of Internal and External Foot Rotation on Peak Knee Adduction Moments and Ankle Moments during Gait in Individuals with Knee Osteoarthritis: A Cross-Sectional Study"

_bioengineering, 2024, doi:10.3390/bioengineering11070696_

Round 1
Reviewer 1 Report
Comments and Suggestions for Authors
The study sought to examine how different foot progression angle (FPA) conditions during unrestricted walking impact the kinetics of the ankle and knee joints in pain-free individuals with knee osteoarthritis, utilizing a 3D motion analysis system featuring two force platforms.
Abstract:
What was the aim of the study?
Method:
• “Thirty-five subjects (9 males and 26 females) diagnosed with knee OA participated.” Who diagnosed the OA, and what was the criterion for having OA in this study?
• “All participants had knee OA without pain.” Please add the definition of their grade of OA based on Kellgren/Lawrence (I, II, and III) in the method.
• “All participants had knee OA without pain.” and “They were recruited from an area orthopedic clinic.” If they did not have pain, why did they visit the orthopedic clinic?
• “Participants had an average age of 62.2 years, a height of 159.3 cm, and a weight of 64.8 kg (Table 1). Average walking speeds in NFP, IFP, and EFP gait trials were 1.24 ± 0.20 m/s, 1.21 ± 0.27 m/s, and 1.23 ± 0.19 m/s, respectively. There were no significant differences in walking speed, cadence, or step length between FPA walking conditions (all p > 0.05).” Move this information to the results section.
• Move Table 1 to the results section.
• Please add a picture that shows the location of reflective markers on a schematic view.
• Did the authors consider or assess the amount of foot hyperpronation or supination, which are confounding factors for OA and walking assessments in the participants?
• Please move Figure 4 to the results section.
• Did the authors consider the effect of hip anteversion angle on their results?
Results:
• In Tables 2 and 4, define "F" as a footnote.
• Why does Table 5 not provide a p-value?
• “Positions * Sides”: If "*" means multiply, use the multiplication sign here.
Discussion:
• What is the generalizability of the results?
• What are the clinical and practical applications of the results?
Comments on the Quality of English LanguageIt requires major revision.
Author Response
Responses to the REVIEWER #1 Comments
The study sought to examine how different foot progression angle (FPA) conditions during unrestricted walking impact the kinetics of the ankle and knee joints in pain-free individuals with knee osteoarthritis, utilizing a 3D motion analysis system featuring two force platforms.
Abstract: What was the aim of the study?
Response: We added the aim of the study in the Abstract.
Method:
• “Thirty-five subjects (9 males and 26 females) diagnosed with knee OA participated.” Who diagnosed the OA, and what was the criterion for having OA in this study?
Response: The diagnosis of knee OA of the participants in this study was performed by an orthopedic surgeon working at a clinic affiliated with this research institution. we added the related contents at line 106-107.
- “All participants had knee OA without pain.” Please add the definition of their grade of OA based on Kellgren/Lawrence (I, II, and III) in the method.
Response: We added the introduction of Kellgren/Lawrence classification at line 103-106 in the Methods section.
- “All participants had knee OA without pain.” and “They were recruited from an area orthopedic clinic.” If they did not have pain, why did they visit the orthopedic clinic?
Response: Patients diagnosed with knee osteoarthritis do not always have knee pain. In addition, most of the participants of this study were classified as K/L grade I or II of knee osteoarthritis and had mild clinical symptoms.
- “Participants had an average age of 62.2 years, a height of 159.3 cm, and a weight of 64.8 kg (Table 1). Average walking speeds in NFP, IFP, and EFP gait trials were 1.24 ± 0.20 m/s, 1.21 ± 0.27 m/s, and 1.23 ± 0.19 m/s, respectively. There were no significant differences in walking speed, cadence, or step length between FPA walking conditions (all p > 0.05).” Move this information to the results section.
Response: In response to this comment, we moved the information to the results section.
- Move Table 1 to the results section.
Response: In response to this comment, we moved the Table 1 to the results section.
- Please add a picture that shows the location of reflective markers on a schematic view.
Response: In response to this comment, we added the Figure 4 as reflective markers setting.
- Did the authors consider or assess the amount of foot hyperpronation or supination, which are confounding factors for OA and walking assessments in the participants?
Response: Although we conducted a walking experiment under the same conditions as possible except for changes in the FPA condition, we were unable to evaluate the effect of foot hyperpronation or supination on the moments of the knee and ankle joints as an exogenous variable of the FPA condition. These details are described in addition to study limitations.
- Please move Figure 4 to the results section.
Response: In response to this comment, we moved the Figure 4.
- Did the authors consider the effect of hip anteversion angle on their results?
Response: Although we conducted a walking experiment under the same conditions as possible except for changes in the FPA condition, we were unable to evaluate the effect of hip anteversion or retroversion on the moments of the knee and ankle joints as an exogenous variable of the FPA condition. These details are described in addition to study limitations.
Results:
- In Tables 2 and 4, define "F" as a footnote.
Response: In response to this comment, we added it in Table 2 and 4.
- Why does Table 5 not provide a p-value?
Response: As already presented in Table 4, there were no significant differences between all peak ankle moment values except ankle inversion moment in the main effect according to the FPA condition. The p values with significant differences are described in the foot note of Table 5.
- “Positions * Sides”: If "*" means multiply, use the multiplication sign here.
Response: In response to this comment, we corrected it.
Discussion:
- What is the generalizability of the results?
Response: In response to this comment, we added the generalizability of the results at line 349-357 in Discussion section.
- What are the clinical and practical applications of the results?
Response: In response to this comment, we added the clinical implication at line 349-357 in Discussion section.
Reviewer 2 Report
Comments and Suggestions for Authors
This manuscript is interesting but failed to reach many aspects of scientific articles.
The values of ankle moments reported in the sagittal plane are not in the range of normative data previously published. This unfortunately makes me doubt about the overall quality of this research as presented in the current version. I strongly suggest the authors to verify the accuracy of their data and to resubmit their work.
Line 36-39
Based on the statement “Knee moments that occur in real time during the stance phase, when the body weight is supported on the ground while walking, are biomechanical variables calculated by multiplying the magnitude of the three-dimensional ground reaction force and the lever arm distance from the knee joint center”
I would like the authors to provide more details on the exact procedure of ankle and knee moment calculation.
Line 95
Abbreviations of NFP, IFP, and EFP have to be described at this stage of the manuscript.
Line 119-120
Use NFP, IFP, and EFP abbreviations.
Line 136
“All biomechanical data were low-pass filtered using a fourth-order Butterworth filter with cutoff frequencies set to 15 Hz and 6 Hz, respectively [22]”.
Ref #22 from Collins et al., 2009, used a 4th order low-pass Butterworth filter (2 pass) with a cut-off frequency set at 6 Hz only. Please provide details for mentioning 15Hz?
Line 146
Quality of Figure 1 should be improved. A zoom of the PDF version provided is not clear enough.
Line 146
The values of intra-individual ankle joint net moments in N.m are not in agreement with the previously published literature (all D.A. Winter papers and more recent Fukui et al., 2016). During walking at self-selected speed, the ankle joint net moments should reach around 1.5 N/kg or around 98 N.m for a 65kg person.
Fukui T, Ueda Y, Kamijo F. Ankle, knee, and hip joint contribution to body support during gait. J Phys Ther Sci. 2016 Oct;28(10):2834-2837. doi: 10.1589/jpts.28.2834. Epub 2016 Oct 28. PMID: 27821945; PMCID: PMC5088136.
Line 290
Table 5. Mean (SD) ankle moment values presented for all three foot progression coditions in stance phase during gait. Spelling “conditions”. This Table should be placed at line 265 and not in the discussion section.
Comments on the Quality of English LanguageThe english quality seems ok
Author Response
Responses to the REVIEWER #2 Comments
This manuscript is interesting but failed to reach many aspects of scientific articles.
The values of ankle moments reported in the sagittal plane are not in the range of normative data previously published. This unfortunately makes me doubt about the overall quality of this research as presented in the current version. I strongly suggest the authors to verify the accuracy of their data and to resubmit their work.
Line 36-39
Based on the statement “Knee moments that occur in real time during the stance phase, when the body weight is supported on the ground while walking, are biomechanical variables calculated by multiplying the magnitude of the three-dimensional ground reaction force and the lever arm distance from the knee joint center”
I would like the authors to provide more details on the exact procedure of ankle and knee moment calculation.
Response: In response to this comment, we added the details at line 160-170 in the Methods section.
Line 95
Abbreviations of NFP, IFP, and EFP have to be described at this stage of the manuscript.
Response: Based on comments from other reviewers, relevant information was moved in the results section.
Line 119-120
Use NFP, IFP, and EFP abbreviations.
Response: Since the contents of Line 95 have been moved to the results, leave it as is.
Line 136
“All biomechanical data were low-pass filtered using a fourth-order Butterworth filter with cutoff frequencies set to 15 Hz and 6 Hz, respectively [22]”.
Ref #22 from Collins et al., 2009, used a 4th order low-pass Butterworth filter (2 pass) with a cut-off frequency set at 6 Hz only. Please provide details for mentioning 15Hz?
Response: As like our previous studies, following 3D motion capture data collection, Visual 3D software was used to analyze kinematic and kinetic data using the Calibrated Anatomical System Technique with a modified oxford foot model. Kinematic data were low-pass filtered with a 4th order Butterworth filter with a cut-off frequency of 6 Hz. Kinetic data were low-pass filtered using a 4th order Butterworth filter with a cut-off frequency of 15 Hz. KAMs were calculated using inverse dynamic analysis. The X-Y-Z Cardan sequence was used to define the order of rotations following the Right Hand Rule about the segment coordinate system axes. Joint kinematic and kinetic data were normalized to the gait cycle starting with initial heel contact. GRF data and joint moments were normalized for body weight. We added detailed information related to this at line 160-167 to the methods section.
Line 146
Quality of Figure 1 should be improved. A zoom of the PDF version provided is not clear enough.
Response: We believe that what you commented on is Figure 2, not Figure 1. So, we added the Figure 2 graph enlarged and modified.
Line 146
The values of intra-individual ankle joint net moments in N.m are not in agreement with the previously published literature (all D.A. Winter papers and more recent Fukui et al., 2016). During walking at self-selected speed, the ankle joint net moments should reach around 1.5 N/kg or around 98 N.m for a 65kg person.
Fukui T, Ueda Y, Kamijo F. Ankle, knee, and hip joint contribution to body support during gait. J Phys Ther Sci. 2016 Oct;28(10):2834-2837. doi: 10.1589/jpts.28.2834. Epub 2016 Oct 28. PMID: 27821945; PMCID: PMC5088136.
Response: In general, it is right to present the moment value occurring in the lower extremity joints during gait in the unit of Nm/kg in the scientific articles. The moment unit presented in this manuscript is Nm∙kg-1 = Nm/kg, but if there is confusion, it will be replaced to Nm/kg. As you mentioned, the ankle moment value presented in the Fukui et al. (2016) paper is similar to that presented in our manuscript, so please check again. Additionally, negative or positive values indicate the direction of motion in a certain motion plane (e.g. plantar flexion (-) or dorsiflexion (+)). The moment values before normalization to the subject's body weight are shown in Figure 2, the ankle plantar flexion net moment was approximately 82 Nm (1.37 Nm/kg) and the dorsiflexion net moment was approximately 12 Nm (0.2 Nm/kg). It was confirmed again that there were no errors in our data processing process and presented ankle moment values. In addition, unfortunately, it is quite difficult to consider the reliability of the Journal of Physical Therapy Science cited above by you as this journal is known to be a predatory journal and has been excluded from many scientific indexes, including SCIE and Scopus.
Line 290
Table 5. Mean (SD) ankle moment values presented for all three foot progression coditions in stance phase during gait. Spelling “conditions”. This Table should be placed at line 265 and not in the discussion section.
Response: In response to this comment, we corrected it and moved the Table 5 to the Results.
Reviewer 3 Report
Comments and Suggestions for Authors
The manuscript addresses an important research question related to the influence of internal and external foot rotation on peak knee adduction moments and ankle moments in individuals with knee OA. However, I have identified some areas where improvements could be made to enhance the clarity and overall presentation of the paper.
1. The title could be more effective by adding more specific information, such as the methodology/design used.
2. Abstract: Without a clear objective statement, the abstract may be confusing or difficult to understand. Readers may be left wondering what the study is about and what the authors are trying to achieve. So, I suggest placing the objective statement at the beginning of the abstract in the context of the background provided.
3. Also, it would be helpful to mention the main implications or practical applications of the study's findings in the conclusion.
4. The introduction provides a good background on knee osteoarthritis and its biomechanical changes. However, it could be expanded to include more recent literature and studies.
5. Consider providing more information on the importance of understanding the relationship between foot rotation and knee joint mechanics in individuals with knee OA.
6. Methods: a detailed description of the study design is imperatively needed. This should enable other researchers to evaluate the study's quality, understand the research process, and make informed judgments about the reliability and generalizability of the findings.
7. The study is not powered enough for the study design and outcome measures. With a relatively small sample size of only 35 participants, it is doubtful whether the study can generate robust and definitive conclusions. A larger sample size would be more appropriate to ensure a higher level of confidence in the study's findings. With increasing the number of participants, the study would be better equipped to capture a more representative range of responses and account for potential variations, thereby enabling researchers to draw more reliable and meaningful conclusions.
8. A sample size of 25 was obtained based on an effect size enough to identify the estimated significance based on a previous study by Hutchison et al. which is not a full-scale study, but rather, a study protocol.
9. The effect size that the authors employed might be very large. So, it yielded a very small sample size, which is seemingly inappropriate for the current study design.
10. The discussion reads well. However, it can be developed better. Authors need to provide a more comprehensive interpretation of the results, compare them with previous research, and discuss their implications.
Author Response
Responses to the REVIEWER #3 Comments
The manuscript addresses an important research question related to the influence of internal and external foot rotation on peak knee adduction moments and ankle moments in individuals with knee OA. However, I have identified some areas where improvements could be made to enhance the clarity and overall presentation of the paper.
- The title could be more effective by adding more specific information, such as the methodology/design used.
Response: In response to this comment, we corrected the title.
- Abstract: Without a clear objective statement, the abstract may be confusing or difficult to understand. Readers may be left wondering what the study is about and what the authors are trying to achieve. So, I suggest placing the objective statement at the beginning of the abstract in the context of the background provided.
Response: We added the aim of the study in the Abstract.
- Also, it would be helpful to mention the main implications or practical applications of the study's findings in the conclusion.
Response: In response to this comment, we added the main clinical implication according to the other reviewer’s comments at line 349-357 in the Discussion section.
- The introduction provides a good background on knee osteoarthritis and its biomechanical changes. However, it could be expanded to include more recent literature and studies.
Response: We added more details at line 55-59 with more recent references in the Introduction section.
- Consider providing more information on the importance of understanding the relationship between foot rotation and knee joint mechanics in individuals with knee OA.
Response: We added more information the relationship between FPA modification walking and knee joint biomechanics at line 68-78 in the Introduction section.
- Methods: a detailed description of the study design is imperatively needed. This should enable other researchers to evaluate the study's quality, understand the research process, and make informed judgments about the reliability and generalizability of the findings.
Response: In response to this comment, we added the details of the study design in methods section.
- The study is not powered enough for the study design and outcome measures. With a relatively small sample size of only 35 participants, it is doubtful whether the study can generate robust and definitive conclusions. A larger sample size would be more appropriate to ensure a higher level of confidence in the study's findings. With increasing the number of participants, the study would be better equipped to capture a more representative range of responses and account for potential variations, thereby enabling researchers to draw more reliable and meaningful conclusions.
Response: We agree with your comments that a large sample size would be better to ensure a high level of confidence of the results. We calculated the sample size using G*Power referring to previous study’s peak KAM values during an immediate-effects study investigating FPA gait modification in individuals with medial knee OA (Simic et al.,2013). The detailed contents were corrected at line 97-100 in the Subject section. Although experimental verification through many subjects reduces type I error and increases the reliability of the results, there are many practical difficulties in conducting research by recruiting a large group of patients with diseases such as knee OA. The number of subjects in this study was 35, but data comparison was performed on 210 moment values of three FPA walking conditions that occurred in both lower extremities, so the amount of data cannot be considered small. The measurement equipment used in this study is a worldwide, highly functional 3D motion analysis system, and the quantitative and objective biomechanical variables obtained through it were suitable for estimating parametric tests.
- A sample size of 25 was obtained based on an effect size enough to identify the estimated significance based on a previous study by Hutchison et al. which is not a full-scale study, but rather, a study protocol.
Response: When we calculated the sample size, we cited the use of Huchison et al.'s method. This was corrected as it could be confusing.
- The effect size that the authors employed might be very large. So, it yielded a very small sample size, which is seemingly inappropriate for the current study design.
Response: There are practical limitations in conducting gait analysis experiment on a large sample of subjects with certain diseases. We added these matters to the study limitations and highlight the need for verification on more patients with knee osteoarthritis in future studies.
- The discussion reads well. However, it can be developed better. Authors need to provide a more comprehensive interpretation of the results, compare them with previous research, and discuss their implications.
Response: In response to this comment, we added a more comprehensive interpretation of the results and the clinical implication at line 336-342, 357-365 in Discussion section.
Reviewer 4 Report
Comments and Suggestions for Authors
This study examined the influence of internal and external foot rotation on peak knee adduction moments and ankle moments during gait in individuals with knee osteoarthritis. This study is intriguing and well executed; however, some improvements in the writing could improve the quality of this manuscript.
Abstract:
1. The abstract is well-written, interesting, and has a lot of significant results.
Introduction:
1. Line 30: Since this study is unrelated to a specific country, I would avoid reporting the data for only one country but worldwide.
2. The hypotheses and study importance can be added after the study aims.
3. Overall, the introduction is well written with clear rationale and aims
Methods:
1. There is a distinct difference between men and women in this study. Please elaborate on how this can affect the study results.
2. Please add the parameters used for G*power.
a. Also, this might better suit the participant’s section.
3. Please add the selected p-level in the statistical analysis.
4. Did you perform any descriptive analyses? Please add.
5. Please consider calculating and reporting effect size for ANOVAs
Results:
1. I would suggest adding some figures, particularly for the mean data (also showing significant results). This will increase the visibility of the results compared to the tables.
2. The table 5 - the descriptive data should be added first.
Discussion:
1. Lines 307-313 are more appropriate for the Introduction. Please elaborate more rather than repeating the results of the previous studies.
a. Same line 323, 331-333
b. This should be mentioned in the Introduction.
2. Lines 334-351 are unnecessarily long. There is a lot of reminding of the aim, results, previous research, and minimal elaboration.
3. Regarding the limitations, see comment 1 for the Methods.
4. Overall, the discussion is long and hard to read, with many reports of previous studies, which is more suitable for the introduction. Please revise the discussion to be a bit more concise where possible.
Author Response
Responses to the REVIEWER #4 Comments
This study examined the influence of internal and external foot rotation on peak knee adduction moments and ankle moments during gait in individuals with knee osteoarthritis. This study is intriguing and well executed; however, some improvements in the writing could improve the quality of this manuscript.
Abstract:
- The abstract is well-written, interesting, and has a lot of significant results.
Introduction:
- Line 30: Since this study is unrelated to a specific country, I would avoid reporting the data for only one country but worldwide.
Response: In response to this comment, we corrected it.
- The hypotheses and study importance can be added after the study aims.
Response: In response to this comment, we added the hypotheses.
- Overall, the introduction is well written with clear rationale and aims
Methods:
- There is a distinct difference between men and women in this study. Please elaborate on how this can affect the study results.
Response: In response to this comment, we added relevant contents at line 381-385 to the Discussion section.
- Please add the parameters used for G*power.
a. Also, this might better suit the participant’s section.
Response: In response to this comment, we added the details of the parameters used for G*power in Participants section.
- Please add the selected p-level in the statistical analysis.
Response: In response to this comment, we added it in statistical analysis section.
- Did you perform any descriptive analyses? Please add.
Response: This study design was a cross-sectional study which focused on the verification of changes in biomechanical variables through gait analysis; therefore, no descriptive analysis was performed.
- Please consider calculating and reporting effect size for ANOVAs
Response: We calculated the sample size and reported effect size using G*Power with previous study’s peak KAM values during an immediate-effects study investigating FPA gait modification in individual with medial knee OA. The detailed contents were corrected at line 96-99 in Subject section. An effect size of 1.0, an alpha level of 0.05, and 90% power were suitable for estimating parametric tests such as ANOVA.
Results:
- I would suggest adding some figures, particularly for the mean data (also showing significant results). This will increase the visibility of the results compared to the tables.
Response: A graph of the average KAM according to FPA condition was presented as Figure 5, and the p value for the only significant variable of ankle peak moment was presented in the foot note of the Table 5.
- The table 5 - the descriptive data should be added first.
Response: Table 5 shows the specific ankle moment average values after conducting a post hoc test because there was a significant difference in the foot position level in Table 4. For visibility, comparison targets with significant differences were added to the foot note of Table 5.
Discussion:
- Lines 307-313 are more appropriate for the Introduction. Please elaborate more rather than repeating the results of the previous studies.
a. Same line 323, 331-333
b. This should be mentioned in the Introduction.
Response: In response to this comment, we moved and corrected the relative contents to line 68-78 in the Introduction section.
- Lines 334-351 are unnecessarily long. There is a lot of reminding of the aim, results, previous research, and minimal elaboration.
Response: In response to this comment, we corrected the contents with minimal elaboration.
- Regarding the limitations, see comment 1 for the Methods.
Response: In response to this comment, we added relevant contents at line 381-385 to the Discussion section.
4. Overall, the discussion is long and hard to read, with many reports of previous studies, which is more suitable for the introduction. Please revise the discussion to be a bit more concise where possible.
Response: In response to this comment, The Discussion section has been overall reduced and corrected.
Round 2
Reviewer 1 Report
Comments and Suggestions for Authors
Thank you for providing the opportunity to review the revised version of the manuscript. The authors have addressed most of my questions. However, I still require further clarification for two of my comments.
1- Line 102: “Their average Kellgren-Lawrence classification was 1.8 ± 0.58 (Grade I: n = 10, Grade II: 102 n = 22, Grade III: n = 3).”
Since three of your patients had Kellgren-Lawrence grade 3, please incorporate the definition of grade 3 into your Kellgren-Lawrence classification in lines 105-106.
2- “Response: Patients diagnosed with knee osteoarthritis do not always have knee pain. In addition, most of the participants of this study were classified as K/L grade I or II of knee osteoarthritis and had mild clinical symptoms.”
Following your response, could you please specify the mild symptoms that necessitate patients to visit the orthopedic surgeon? Kindly include them in the method section.
Author Response
Responses to the REVIEWER #1 Comments
Thank you for providing the opportunity to review the revised version of the manuscript. The authors have addressed most of my questions. However, I still require further clarification for two of my comments.
1- Line 102: “Their average Kellgren-Lawrence classification was 1.8 ± 0.58 (Grade I: n = 10, Grade II: 102 n = 22, Grade III: n = 3).”
Since three of your patients had Kellgren-Lawrence grade 3, please incorporate the definition of grade 3 into your Kellgren-Lawrence classification in lines 105-106.
Response: In response to this comment, we added the definition of K/L grade III.
2- “Response: Patients diagnosed with knee osteoarthritis do not always have knee pain. In addition, most of the participants of this study were classified as K/L grade I or II of knee osteoarthritis and had mild clinical symptoms.”
Following your response, could you please specify the mild symptoms that necessitate patients to visit the orthopedic surgeon? Kindly include them in the method section.
Response: In response to this comment, we added the related contents in the Subjects section.
Reviewer 2 Report
Comments and Suggestions for Authors
The authors made the corrections of the manuscript and provide answers to my questions.
Figure 2 is still not sharp (at least with my PDF viewer). That's why I was not able to see the numbers.
Line 8: The aim of the study was to verify effects of the foot progression angle (FPA) modification during walking on moments of the ankle and knee joints in individuals with knee OA.
Author Response
Responses to the REVIEWER #2 Comments
The authors made the corrections of the manuscript and provide answers to my questions.
Figure 2 is still not sharp (at least with my PDF viewer). That's why I was not able to see the numbers.
Response: We re-created the graph in Figure 2 to identify the numbers.
Line 8: The aim of the study was to verify effects of the foot progression angle (FPA) modification during walking on moments of the ankle and knee joints in individuals with knee OA.
Response: We corrected it.
Reviewer 3 Report
Comments and Suggestions for Authors
The authors have effectively addressed the previous comments and suggestions to a satisfactory extent. From my assessment, I believe the manuscript is ready for publication in its current form.
Author Response
Responses to the REVIEWER #3 Comments
The authors have effectively addressed the previous comments and suggestions to a satisfactory extent. From my assessment, I believe the manuscript is ready for publication in its current form.
Response: Thank you for reviewing our manuscript.
Reviewer 4 Report
Comments and Suggestions for Authors
- Did you perform any descriptive analyses? Please add.
Response: This study design was a cross-sectional study which focused on the verification of changes in biomechanical variables through gait analysis; therefore, no descriptive analysis was performed.
The authors report mean/averaged values throughout the manuscript, including in Table 5. Therefore, you did some descriptive analysis. Which one, mean values, standard deviation? Please add that to the methods.
- Please consider calculating and reporting effect size for ANOVAs
Response: We calculated the sample size and reported effect size using G*Power with previous study’s peak KAM values during an immediate-effects study investigating FPA gait modification in individual with medial knee OA. The detailed contents were corrected at line 96-99 in Subject section. An effect size of 1.0, an alpha level of 0.05, and 90% power were suitable for estimating parametric tests such as ANOVA.
This issue is not about G*power but the effect size of the ANOVA performed https://lbecker.uccs.edu/glm_effectsize. My suggestion is to calculate the effect size for ANOVA and report it. Also, don’t forget to add it to the statistical analysis.
Author Response
Responses to the REVIEWER #4 Comments
- Did you perform any descriptive analyses? Please add.
Response: This study design was a cross-sectional study which focused on the verification of changes in biomechanical variables through gait analysis; therefore, no descriptive analysis was performed.
The authors report mean/averaged values throughout the manuscript, including in Table 5. Therefore, you did some descriptive analysis. Which one, mean values, standard deviation? Please add that to the methods.
Response: In response to this comment, we added it to the Statistical analysis section.
- Please consider calculating and reporting effect size for ANOVAs
Response: We calculated the sample size and reported effect size using G*Power with previous study’s peak KAM values during an immediate-effects study investigating FPA gait modification in individual with medial knee OA. The detailed contents were corrected at line 96-99 in Subject section. An effect size of 1.0, an alpha level of 0.05, and 90% power were suitable for estimating parametric tests such as ANOVA.
This issue is not about G*power but the effect size of the ANOVA performed https://lbecker.uccs.edu/glm_effectsize. My suggestion is to calculate the effect size for ANOVA and report it. Also, don’t forget to add it to the statistical analysis.
Response: In response to this comment, we added the related contents to the Statistical analysis and Results sections.